

# On Turbulence Models and LiDAR Measurements for Wind Turbine Control

Liang Dong[1], Wai Hou Lio[1], and Eric Simley[2]

[1]Department of Wind Energy, Technical University of Denmark (DTU), Frederiksborgvej 399, 4000, Roskilde, Denmark
[2]National Renewable Energy Laboratory (NREL), 15013 Denver West Parkway, Golden, Colorado 80401, USA

**Correspondence:** Liang Dong (ldong@dtu.dk)

**Abstract.** To provide comprehensive information that will assist in making decisions regarding the adoption of LiDAR assisted control (LAC) in wind turbine design, this paper investigates the impact of different turbulence models on the coherence between the rotor effective wind speed and LiDAR measurement. First, the differences between the Kaimal and Mann models are discussed, including the power spectrum and spatial coherence. Next, two types of LiDAR systems are examined to analyze the LiDAR measurement coherence based on commercially available LiDAR scan patterns. Finally, numerical simulations have been performed to compare the LiDAR measurement coherence for different rotor sizes. This work confirms the association between the measurement coherence and the turbulence model. The results indicate that the LiDAR measurement coherence with the Mann turbulence model is lower than that with the Kaimal turbulence model. In other words, the value creation of LAC, evaluated using the Kaimal turbulence model, will be diminished if the Mann turbulence model is used instead. In particular, the difference in coherence is more significant for larger rotors. As a result, this paper suggests that the impacts of different turbulence models should be considered as uncertainties while evaluating the benefits of LAC.

## 1 Introduction

Turbine-mounted LiDAR sensors provide preview information about the inflow wind to be used for improving wind turbine control, which is referred to as wind turbine integrated LiDAR assisted control (LAC). LAC is a promising technology for reducing wind turbine loads and the levelized cost of energy (LCOE) (Scholbrock et al., 2016; Simley et al., 2020). The potential benefits have been demonstrated in several works by simulation (Schlipf et al., 2010; Bossanyi, 2013; Schlipf et al., 2013b; Bossanyi et al., 2014) as well as field experiments (Kumar et al., 2015; Fleming et al., 2014; Schlipf et al., 2014).

The topic of the optimal LiDAR scan pattern for wind energy applications is critical for the widespread deployment of LAC. Both practical considerations for overcoming the obstacles of LAC application and for optimizing LiDAR scan patterns were discussed in an International Energy Agency (IEA) Wind Task 32 workshop (Simley et al., 2018). The correlation between the rotor effective wind speeds measured by the LiDAR and experienced by the rotor has been discussed in Haizmann et al. (2015), Simley et al. (2012), and Schlipf et al. (2013a), in which the magnitude-squared coherence is suggested as a useful metric to quantify LiDAR measurement quality. A fundamental component of LiDAR measurement coherence is the theoretical spatial coherence of the turbulence: (a) the transverse and vertical spatial coherence is defined in the International Electrotechnical





Commission (IEC) design standard (IEC, 2019); (b) wind evolution models (Bossanyi, 2013; Simley and Pao, 2015) are defined in terms of longitudinal spatial coherence. Note that the actual coherence in the field could be different from the theoretical coherence, thus experimental validation by field testing is important as well.

Although extensive research has been carried out on evaluating LiDAR measurement coherence, there is a clear knowledge gap regarding the impact of turbulence models on the LiDAR measurement coherence. The wind field model used in most of
the above-mentioned studies consists of the Kaimal turbulence spectrum and the spatial coherence model defined in the IEC standard (IEC, 2019). However, there are two different turbulence models defined in the IEC standard: the Mann turbulence model and the Kaimal turbulence model. The impact of different wind fields on the dynamic response of an offshore wind turbine has been evaluated by Nybø et al. (2020). Held and Mann (2019) extended the previous works by Haizmann et al. (2015), Simley et al. (2012), and Schlipf et al. (2013a) to analyze LiDAR measurement coherence with both the Mann turbulence
model and Kaimal turbulence model. The theoretical coherence results were compared to field data from a nacelle LiDAR mounted on a Vestas V52 wind turbine. The results showed that the experimental data fit better to the coherence predicted by the Mann turbulence model, and the prediction based on the Kaimal turbulence model underestimates the coherence. However, the coherence analysis focused solely on a turbine with a small rotor diameter of 52 m; the impact of different rotor sizes and LiDAR scan patterns on coherence have not been investigated in the work (Held and Mann, 2019).

With the advent of larger rotor sizes and more flexible wind turbines, evaluating the value creation of LAC is becoming increasingly important. The analysis in this work is based on the framework proposed by Simley et al. (2018) and Held and Mann (2019). The specific objective of this study is to investigate the impact of different turbulence models on the LiDAR measurement coherence, especially for large rotor sizes (i.e., the Technical University of Denmark (DTU) 10-MW reference turbine with a rotor diameter of 178 m (Bak et al., 2013)), whereby the analysis can shed light on how to reasonably evaluate
LAC benefits. First, the differences between the Kaimal and Mann models are discussed. Then two types of commercial continuous wave (CW) LiDAR systems are examined to analyze the LiDAR measurement coherence, including a 4-beam LiDAR and 50-beam circular scan LiDAR. The LiDAR measurement model has been created based on work by Simley et al. (2011) and numerical simulations have been performed to compare the LiDAR measurement coherence.

The remainder of this paper is organized as follows: Section 2 briefly describes the different turbulence models and compares
the power spectra. The LiDAR measurement model is established in Section 3. In Section 4, numerical simulations for different LiDAR scan patterns and rotor sizes are performed. The conclusions and suggestions for future work are summarized in Section 5.

## 2  Preliminaries and evaluation of different turbulence models

Two different turbulence models are commonly used to evaluate the design loads in the IEC standard (IEC, 2019): the Kaimal
spectrum with exponential coherence model (Kaimal model) and the Mann turbulence model (Mann model). The turbulence models use similar power spectra, and the major difference is the spatial distribution of the wind velocities.



## 2.1 Kaimal model

The advantage of the Kaimal model is that the one-dimensional spectra are expressed as simple analytic expressions. The wind disturbance is described as turbulent velocity fluctuations, and is assumed to be a stationary and random vector field with zero-mean Gaussian statistics. The power spectral densities (PSD) of each wind components are given in non-dimensional form:

$$\frac{f S_{\text{k}}(f)}{\sigma_{\text{k}}^2} = \frac{4 f L_{\text{k}}/V_{\text{hub}}}{(1 + 6 f L_{\text{k}}/V_{\text{hub}})^{5/3}},\tag{1}$$

where $f$ is the frequency in Hertz, while the subscript k denotes the index of the velocity component in the longitudinal $u$, lateral $v$, and upward $w$ direction, respectively. The single-sided velocity component spectrum is denoted as $S_{\text{k}}$, while $\sigma_{\text{k}}$ and $L_{\text{k}}$ represent the standard deviation and integral length scale parameters of the velocity component, respectively. The wind speed at hub height is denoted as $V_{\text{hub}}$.

For the longitudinal velocity component $u$, $\sigma_u$ is the representative value of the turbulence standard deviation, and $L_{\text{u}}$ is defined as $L_u = 8.1 \Lambda_u$. For a modern wind turbine, the hub height is typically above $z \geq 60$ m and the longitudinal length scale parameter is $\Lambda_u = 42$ m.

The cross power spectral density (CPSD) $S_{u_i,u_j}(f)$ between the wind at two spatially separated points $u_i, u_j$ can be determined from the definition of spatial co-coherence $\gamma_{i,j}$:

$$\gamma_{i,j}(f) = \Re\left(\frac{S_{u_i,u_j}}{\sqrt{S_{u_i,u_i} S_{u_j,u_j}}}\right),\tag{2}$$

where $S_{u_i,u_i}$ and $S_{u_j,u_j}$ are the PSDs of the wind speed at two different locations, $i$ and $j$. The symbol $\Re$ denotes the real part of a complex number. Please note that the coherence can be split into a real part and an imaginary part, which are referred to as co-coherence and quad-coherence (Nybø et al., 2020). The coherence expressed in Eq. (2) is in the real part form.

According to the IEC standard (IEC, 2019), the following exponential coherence model can be used in conjunction with the Kaimal PSD:

$$\gamma_{i,j}(f) = \exp\left[-12\left(\left(\frac{fr}{V_{\text{hub}}}\right)^2 + \left(\frac{0.12 r}{L_c}\right)^2\right)^{0.5}\right],\tag{3}$$

where $r$ is the magnitude of the distance between the two points projected onto a plane normal to the averaged wind direction and $L_c = L_u$ is the coherence scale parameter. The definition in Eq. (3) ignores the quad-coherence, thus the wind velocity fluctuations are assumed to be in the same phase. This assumption may be reasonable for small rotor sizes, but can be questioned for larger rotor sizes (Eliassen and Obhrai, 2016).

## 2.2 Mann model

The Mann turbulence model (Mann, 1994) is a spectral tensor model based on von Karman's model, which combines rapid distortion theory (RDT) with considerations about eddy lifetimes. The RDT in the Mann model gives an equation for the



evolution or the "stretching" of the spectral tensor, and the tensor will be more and more "anisotropic" with time. RDT will finally influence the transverse-vertical coherence in the rotor plane.

The three-dimensional wind field can be represented by the vector field

$$\mathbf{u}(\mathbf{x}) = (u_1, u_2, u_3) = (u, v, w). \tag{4}$$

Because of homogeneity, the covariance tensor is a function of the separation vector $\mathbf{r}$ between two points, and is defined as follows:

$$R_{ij}((r)) = \langle u_i(\mathbf{x})u_j(\mathbf{x}+\mathbf{r})\rangle, \tag{5}$$

where $\langle \; \rangle$ denotes ensemble averaging.

All second order statistics of turbulence, such as variances and cross spectra, can be derived from the covariance tensor. The
spectral tensor is given by:

$$\Phi_{ij}(\mathbf{k}) = \frac{1}{(2\pi)^3} \int R_{ij}(\mathbf{r})e^{-i\mathbf{k}\cdot\mathbf{r}}\mathrm{d}\mathbf{r}, \tag{6}$$

where $\int \mathrm{d}\mathbf{r} = \int_{-\infty}^{\infty}\int_{-\infty}^{\infty}\int_{-\infty}^{\infty} \mathrm{d}r_1\mathrm{d}r_2\mathrm{d}r_3$, $\mathbf{k} = (k_1, k_2, k_3)$ is the non-dimensional spatial wave number for the three component directions, $k = 2\pi f/\bar{U}$, and $\bar{U}$ is the mean wind speed. The resulting spectral tensor components can be found in Annex C of the IEC standard (IEC, 2019).

For three-dimensional turbulent velocity vector $\mathbf{u}(\mathbf{x})$, the velocity components are determined from a decomposition of the spectral tensor and an approximation by discrete Fourier transform, following the procedure detailed in Mann (1998). Compared to the Kaimal spectrum and exponential coherence model, the advantage of using the Mann model to analyze LiDAR measurements is that it provides a three-dimensional spectral tensor. The Mann model includes correlation between the $(u, v, w)$ components, whereas the Kaimal model has no correlation between different wind components.

The Mann model is based on three adjustable parameters: $\alpha\epsilon^{2/3}$, the Kolmogorov constant multiplied with the rate of the viscous dissipation of specific turbulent kinetic energy raised to the power of two-thirds, the length scale $l$, and the non-dimensional parameter $\Gamma$ related to the lifetime of the eddies.

The co-coherence $\gamma_{ij}$ for the spatial separations (grid point $i$ and $j$) normal to the longitudinal direction is defined as

$$\gamma_{ij}(k_1, \Delta_y, \Delta_z) = \Re\left(\frac{\int_{-\infty}^{\infty}\int_{-\infty}^{\infty}\Phi_{ij}(\mathbf{k})e^{ik_2\Delta_y}e^{ik_3\Delta_z}\mathrm{d}k_2\mathrm{d}k_3}{\sqrt{\Psi_{ii}(k_1)\Psi_{jj}(k_1)}}\right), \tag{7}$$

where $\Delta_y$ is the lateral separation distance and $\Delta_z$ is the vertical separation distance. When the two indices $i = j$, then $\Delta_y = \Delta_z = 0$ and the wave number auto-spectrum $\Psi_{ii}(k_1)$ is expressed as

$$\Psi_{ii}(k_1) = \int\limits_{-\infty}^{\infty}\int\limits_{-\infty}^{\infty}\Phi_{ii}(\mathbf{k})\mathrm{d}k_2\mathrm{d}k_3. \tag{8}$$





### 2.3 Evaluation using different turbulence generators

The theoretical turbulence models are quite complicated, especially for the Mann model, although the application of the Mann
model only requires three parameters ($\alpha\epsilon^{2/3}$, $l$, $\Gamma$). Therefore, numerical simulations have been performed to compare the
different turbulence models in this work.

#### 2.3.1 Coordinate system

The coordinate system of the wind box as well as the LiDAR scan patterns is shown in Fig. 1. The size of the wind box should
cover the entire rotor disc. The directions of the wind components $(u, v, w)$ are aligned with the directions of the coordinate
system axes $(x, y, z)$. The LiDAR scan pattern will be elaborated in Section 3.2.

#### 2.3.2 Turbulence generator

To generate the wind box for further analysis, two different turbulence simulators are used. The Kaimal model can be generated
using the turbulence simulator TurbSim (Jonkman and Buhl Jr., 2006), while the Mann model is generated by HAWC2 (Hansen
et al., 2018).
All numerical simulations are performed for a wind field with mean wind speed $\bar{U} = 12$ m/s and turbulence intensity given
by the IEC Class A normal turbulence model (NTM). The parameters of the three-dimensional wind box are listed in Table 1.
The grid size in the vertical and lateral directions is defined by the size of the wind box $L_{\mathrm{grid}}$ and number of grid points $N_{\mathrm{grids}}$.
Assuming Taylor's hypothesis of frozen turbulence, the grid size along the mean wind direction is defined as $\bar{U}T/N_x$, where
$\bar{U}$ is the mean wind speed, $T$ is the total time, and $N_x$ is the number of longitudinal grid points.
Since the Mann turbulence fields are normally re-scaled to the specified turbulence intensity inside HAWC2, the parameter
$\alpha\epsilon^{2/3}$ is chosen to be 1 and the shear parameter $\Gamma$ should be approximately 3.9 for neutral conditions. The length scale $l$ is
recommended to be $l = 0.7\Lambda_u$ for normal conditions.

   The method used in TurbSim is the Veer's approach (Veers, 1988) wherein the PSDs in Eq. (1) and coherence function
in Eq. (3) are used to correlate the Fourier components of different points in the $y - z$ plane. Then the inverse fast Fourier
transform (IFFT) is applied to obtain the correlated time series at each grid point. Although in the IEC standard the coherence
function is only applied to the $u$ component, the Veer's approach is extended to apply the coherence to the $(v, w)$ components in
this work as well. It is assumed that the spatial coherence formula presented in Eq. (3) applies to all wind components $(u, v, w)$,
and the length scales for the different components are the same as defined for the PSDs. Otherwise, the LiDAR measurement
error could be unrealistically low. In contrast, the Mann model creates a turbulence field that is fully correlated in the $(x, y, z)$
directions.

#### 2.3.3 Turbulence spectrum comparison

The differences between the Mann and Kaimal models are discussed in this section.

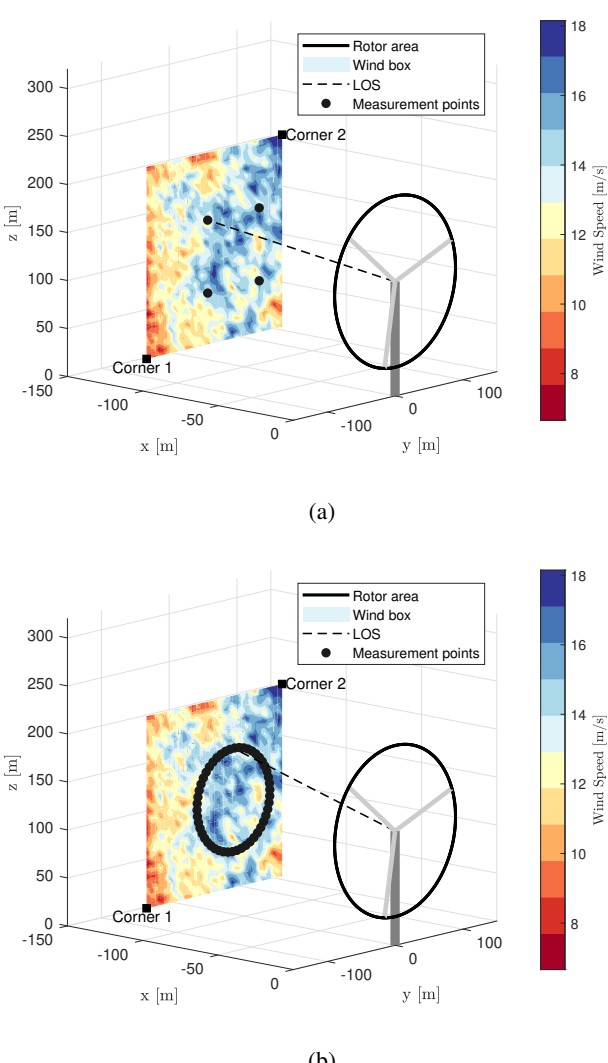

(a)

(b)

**Figure 1.** Coordinate system of wind box and LiDAR scan patterns. The wind box is shown using the color map. Two corners are marked as black squares (Corner 1 and Corner 2). Two commercial CW LiDAR scan patterns are shown: (a) 4-beam CW LiDAR; (b) 50-beam circular scan CW LiDAR. The dashed line represents the line-of-sight direction.





**Table 1.** Settings for generating the turbulence box.

| Symbol | Description | Value | Unit |
|---|---|---|---|
| $T$ | length of time series | 600 | s |
| $t_s$ | sampling time | 0.05 | s |
| $z_h$ | center height of the grid | 119 | m |
| $L_{\mathrm{grid}}$ | width and height of the wind box | 200 | m |
| $\alpha$ | power law wind shear exponent | 0.2 | - |
| $\alpha_v$ | vertical inflow angle | 0 | deg |
| $I_{\mathrm{ref}}$ | reference turbulence intensity | 0.16 (Class A, NTM) | |
| $\bar{U}$ | mean wind speed | 12 | m/s |
| $N_{\mathrm{grids}}$ | number of grid points | 32 | - |
| $N_{\mathrm{x}}$ | number of longitudinal grid points | 8192 | - |

Fig. 2 shows the theoretical co-coherence $\gamma_{i,j}$ at different separation distances, in which the lateral separation distance $\Delta_y$ and vertical separation distance $\Delta_z$ are selected to be 10 m, 30 m, and 50 m. Some interesting findings are:

1. A clear trend can be seen in Fig. 2a wherein the lateral co-coherence reduces as the lateral separation distance increases. With the small separation distance 10 m, the coherence with the Mann model is higher than with the Kaimal model. Conversely, with increasing separation distance, the co-coherence with the Mann model falls sharply compared with the co-coherence with the Kaimal model; the co-coherence with the Mann model is far below the co-coherence with the Kaimal model for $\Delta_y = 50$ m.

2. For vertical separations in Fig. 2b, the co-coherence with the Mann model is always higher than that with the Kaimal model for low wave numbers. Unlike the lateral co-coherence, the vertical co-coherence does not drastically decrease with increasing separation distance.

3. The co-coherence with the Mann model is negative in some frequency ranges, which is not the case for the exponential coherence model with the Kaimal model expressed in Eq. (3). This implies an opposite phase of the wind components for
some frequencies. Chougule et al. (2012) investigated the vertical cross-spectral phases in neutral atmospheric flow; the work demonstrated that the phase angle of the wind component $u$ increases with stream-wise wave number and vertical separation distance.

With the advent of larger rotor sizes, LiDAR measurements must scan a larger area upstream of the rotor. So the findings above indicate that the choice of turbulence model strongly influences the coherence of LiDAR measurements. This impact
should be considered while evaluating the benefits of LAC.



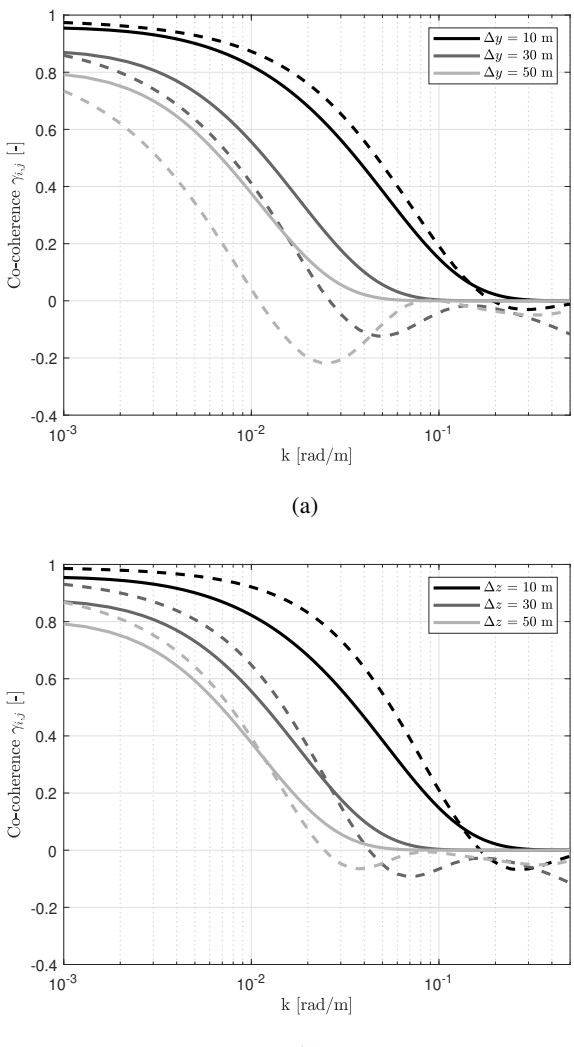

(a)

(b)

**Figure 2.** Co-coherence at different separation distances. $\Delta_y$ and $\Delta_z$ represent the lateral and vertical separation distances, respectively. (a) Lateral co-coherence, $\Delta_z = 0$ m; (b) Vertical co-coherence, $\Delta_y = 0$ m. Dashed lines denote the Mann model and solid lines denote the Kaimal model.





## 3 Modeling of LiDAR wind speed measurements

### 3.1 LiDAR coordinate system

Two different scan patterns based on commercial nacelle-mounted LiDARs are investigated here to illustrate the impact of different turbulence models on LiDAR measurement quality: a 4-beam scan pattern (Fig. 1a) and a 50-beam circular scan pattern (Fig. 1b). The LiDAR is mounted on the nacelle and the scan pattern may contain many different measurement points as shown in Fig. 1. Each scan pattern is further defined by the upstream preview distance $d$ in the $x$ direction and radial distance $r$ between the scan point and the hub center in $y-z$ plane. The LiDAR is assumed to be installed at the hub center for simplicity.

As suggested by Simley et al. (2018), the optimal LiDAR scan radius and preview distance used to achieve the best representation of the actual wind variables of interest that interact with the turbine can be expressed in non-dimensional units relative to the rotor radius. Coherence bandwidth is commonly used as a performance metric for LAC, and will be described in detail in Section 4.2. The optimal scan parameters for maximizing the coherence bandwidth are summarized in Table 2 and the LiDAR scan parameters are defined accordingly in this work.

### 3.2 LiDAR simulator

The line-of-sight (LOS) velocity from a LiDAR system can be expressed as:

$$v_{LOS} = -l_x u - l_y v - l_z w, \tag{9}$$

where $l = [l_x, l_y, l_z]$ denotes the unit vector in the direction that the beam is oriented. Note that the sign of the upwind direction is negative.

The true velocity measured by a LiDAR is a spatial average of the LOS velocities along the LiDAR beam, which is described by the range weighting function. The range weighting function for continuous-wave LiDARs is expressed as follows (Simley et al., 2014):

$$W_L(F, \Delta) = \frac{K_N}{\Delta^2 + (1 - \frac{\Delta}{F})^2 R_R^2}, \tag{10}$$

where $F$ denotes the LiDAR focal distance, $\Delta$ denotes the distance from the focus position along the beam direction, $K_N$ is a normalizing factor so that the integral of $W_L$ from $-\infty$ to $\infty$ gives unity. $R_R$ is the Rayleigh range and is given by:

$$R_R = \frac{\pi a_2^2}{\lambda}, \tag{11}$$

where $\lambda$ is the laser wavelength and $a_2$ is the beam radius at the output lens, which is calculated for the point at which the intensity has dropped to $e^{-2}$ of its value at the beam centre. The LiDAR beam radius $a_2$ takes the value 28 mm, which is broadly equivalent to the beam radius for current commercial LiDAR products (Pena et al., 2015). The wavelength $\lambda$ is assumed to be the telecommunications wavelength of $1.55 \times 10^{-6}$ m. More details regarding LiDAR modeling can be found in Simley et al. (2014).



**Table 2.** The optimal LiDAR scan parameters for maximizing coherence bandwidth. Optimal scan radii $r$ and preview distances $d$ are expressed in non-dimensional units normalized by the rotor radius $R$. The parameters are chosen according to work by Simley et al. (2018).

| Symbol | Description | 4-Beam CW | Circular Scan | Unit |
|--------|-------------|-----------|---------------|------|
| $r$ | scan radius | $0.5R$ | $0.6R$ | - |
| $d$ | preview distance | $1.2R$ | $1.2R$ | - |
| $\theta$ | cone angle of LOS beam | 22.6 | 26.6 | deg |

### 3.3 Rotor effective wind speed reconstruction

The rotor effective wind speed (REWS) is commonly used to indicate the rotor averaged wind condition. The REWS is modeled as a sum of the $u$ component wind speeds across the entire rotor disk area, assuming $N_p$ points on the rotor disk:

$$u_{\text{eff}} = \frac{1}{N_p} \sum_{i=1}^{N_p} u_i. \tag{12}$$

The method of reconstructing the rotor effective wind speed from LiDAR measurements has been discussed by Schlipf et al. (2011). The LiDAR can only measure the wind speed component along the LOS; therefore, at least three beams are needed to estimate the three-dimensional wind vector at a single point. This limitation is referred to as the cyclops dilemma (Schlipf et al., 2011). Due to the cyclops dilemma and for the purpose of collective blade pitch control, the most common assumptions for reconstructing wind speeds from LiDAR measurements are:

1. no vertical $w$ and lateral $v$ wind component,

2. no shears or inflow angles.

The solution for estimating the rotor effective wind speed from LOS measurements is given by

$$u_{\text{lid}} = -\frac{1}{N} \sum_{i=1}^{N} \frac{v_{los,i}}{l_{x,i}}, \tag{13}$$

where $N$ denotes the number of unique beams and $l_{x,i}$ denotes the $x$ component of the orientation of beam $i$. The wind speed estimate $u_{\text{lid}}$ represents the average wind speed for a LiDAR measuring $N$ points upstream of the turbine.

## 4 Influence of different turbulence models on LiDAR measurement coherence

### 4.1 Numerical simulation settings

In order to investigate the impact of different turbulence models on LiDAR measurement coherence, numerical simulations have been performed. Apart from the Vestas V52 with a 52-m rotor diameter, two other reference wind turbines are used, including the National Renewable Energy Laboratory (NREL) 5-MW reference turbine with a 126-m rotor diameter (Jonkman





et al., 2009) and the DTU 10-MW reference turbine with a 178-m rotor diameter. These two rotor sizes represent typical values
for onshore and offshore turbines, respectively.

The numerical simulations include 18 random turbulence boxes with different seeds for each turbulence model. The simulation time is 600 s. Therefore, the combination of two types of LiDARs, three different rotor sizes, and two turbulence models results in 12 separate scenarios, and 18 random realizations for each scenario.

### 4.2   Criteria for evaluating measurement quality and benefits

For indicating the measurement quality, the wave number $k$ at which the magnitude-squared coherence $\gamma^2$ between $u_{\mathrm{lid}}$ in
Eq. (13) and $u_{\mathrm{eff}}$ in Eq. (12) drops below 0.5 is commonly used as a performance metric (Schlipf et al., 2013b). This metric
is referred to as the coherence bandwidth $k_{0.5}$ in this work. The wave number $k = 2\pi/L$ is the inverse of the eddy diameter,
where the integral length scale $L$ is representative of the eddy size at a particular location. So the smallest detectable eddy size
measured by a LiDAR is defined by the wave number $k_{0.5}$. In other words, the smallest detectable eddy can be interpreted as
the eddy size that can be captured with a correlation of 50%.

The eddy can be assumed to be a three-dimensional spherical structure, which will move along the mean wind flow direction
and eventually interact with the turbine rotor. The thrust load induced by a $1D$-diameter eddy across the rotor in the lateral and
vertical directions can be mitigated by pitching the blades to feather. In addition, the eddy size in the longitudinal direction is
inversely proportional to the frequency at which the eddy interacts with the turbine, which in turn drives the required control
system bandwidth needed to respond to the wind disturbance. For LiDAR assisted collective pitch control, if the LiDAR can
accurately capture the trend of $1D$-diameter eddies, then the pitch action can effectively reduce the thrust variation. Thus, the
magnitude-squared coherence $\gamma^2$ at $k = 2\pi/D$ is the most critical metric.

By optimizing the LiDAR scan pattern, the highest measurement coherence bandwidth can be achieved, but the cost of
LiDAR will increase as well. Meanwhile, the benefits of fatigue load reduction may reach a plateau. Generally speaking,
the lower the value of $k_{0.5}$, the lower the LAC benefits. Integrating LAC into the turbine design phase involves a trade-off
optimization problem to consider the turbine cost and LiDAR cost simultaneously.

### 4.3   Coherence analysis

Based on the simulation results, the magnitude-squared coherence $\gamma^2$ between the LiDAR measurements and rotor effective
wind speeds are presented in Fig. 3 for the different scenarios investigated. For brevity, the dash-dot line labeled as $1D$ represents the wave number corresponding to the rotor diameter $D$, whereas $2D$ indicates the wave number corresponding to
two rotor diameters. It can be clearly seen that the 50-beam circular scan LiDAR can achieve higher measurement coherence
compared to the 4-beam LiDAR. For the NREL 5-MW turbine and the Kaimal model (see Fig. 3 (c) - (d)), the maximum
coherence bandwidth $k_{0.5}$ is approximately 0.03 rad/m for the 4-beam scan and 0.05 rad/s for the 50-beam scan. These results
corroborate the findings of previous work by Simley et al. (2018).

The key findings of this study are included in the following discussion. For brevity, the magnitude-squared coherence with
the Mann model is represented by $\gamma^2_{\mathrm{Mann}}$ and the magnitude-squared coherence with the Kaimal model is represented by



$\gamma^2_{\text{Kaimal}}$. Corresponding theoretical coherence curves are also included in this figure following methods described in work by Held and Mann (2019) and Schlipf et al. (2013a).

1. For the Vestas V52 turbine in Fig. 3 (a) - (b), $\gamma^2_{\text{Mann}}$ is higher than $\gamma^2_{\text{Kaimal}}$ in the low wave number region $k \leq 0.06$ rad/m, which aligns with the findings of the work by Held and Mann (2019), in which the authors suggested that the Kaimal model gave a slight underestimation of the measurement coherence for a 52-m rotor diameter, and the coherence predicted from the Kaimal model is lower than the coherence predicted from the Mann model.

2. For the NREL 5-MW turbine in Fig. 3 (c) - (d), $\gamma^2_{\text{Mann}}$ is slightly higher than $\gamma^2_{\text{Kaimal}}$ for low wave numbers. Then, the
coherence starts to separate around $2D$. Specifically, $\gamma^2_{\text{Mann}}$ decreases more sharply than $\gamma^2_{\text{Kaimal}}$ when $k$ exceeds $2D$.

3. For the DTU 10-MW turbine in Fig. 3 (e) - (f), the trend follows the trend with the NREL 5-MW turbine, but $\gamma^2_{\text{Mann}}$ is considerably lower than $\gamma^2_{\text{Kaimal}}$. The coherence $\gamma^2_{\text{Mann}}$ drastically decreases before $2D$. For increasing wave numbers, larger discrepancies are noticeable between $\gamma^2_{\text{Mann}}$ and $\gamma^2_{\text{Kaimal}}$.

4. The additional measurement points with the circular scan provide an obvious improvement in measurement coherence
in the frequency band $k \in [\frac{2\pi}{2D}, \frac{2\pi}{1D}]$. The maximum coherence bandwidth $k_{0.5}$ can reach $1D$ with the 50-beam circular scan. The Kaimal model indicates that the 50-beam circular scan is a better scan pattern and can lead to realizing the full potential benefits of LiDAR assisted collective pitch control. Surprisingly, the maximum coherence bandwidth $k_{0.5}$ with the Mann model is far below $1D$, which will lead to lower benefits.

A novel finding in this work is that the coherence with the Mann model is lower that that with the Kaimal model, which
has not previously been found in the literature. These results are in accord with the theoretical coherence shown in Fig. 2, indicating lower coherence with the Mann model for larger separation distances. In summary, these results provide important insights into the impact of different turbulence models on LiDAR measurement coherence. The differences between $\gamma^2_{\text{Mann}}$ and $\gamma^2_{\text{Kaimal}}$ are significant. If the wind conditions at a site agree with the Mann model, the lower coherence with the Mann model will diminish the advantages of LAC because inappropriate blade pitch actions in response to the LiDAR measurements
will deteriorate the turbine structural loading. It can therefore be suggested that the turbulence model needs to be carefully considered while integrating the LAC solution with larger-rotor turbine designs.

## 5   Conclusions

This work confirms the association between LiDAR measurement coherence and the turbulence model. Our results suggest that this impact should be considered as an uncertainty when evaluating the benefits of LAC. Note that the impacts on the
load reduction need to be further investigated. More broadly, research is also needed to determine which kind of LiDAR is most suitable for site-specific atmospheric conditions. Further research should be undertaken to provide guidelines on how to determine the optimal scan pattern for different turbulence conditions and rotor sizes.



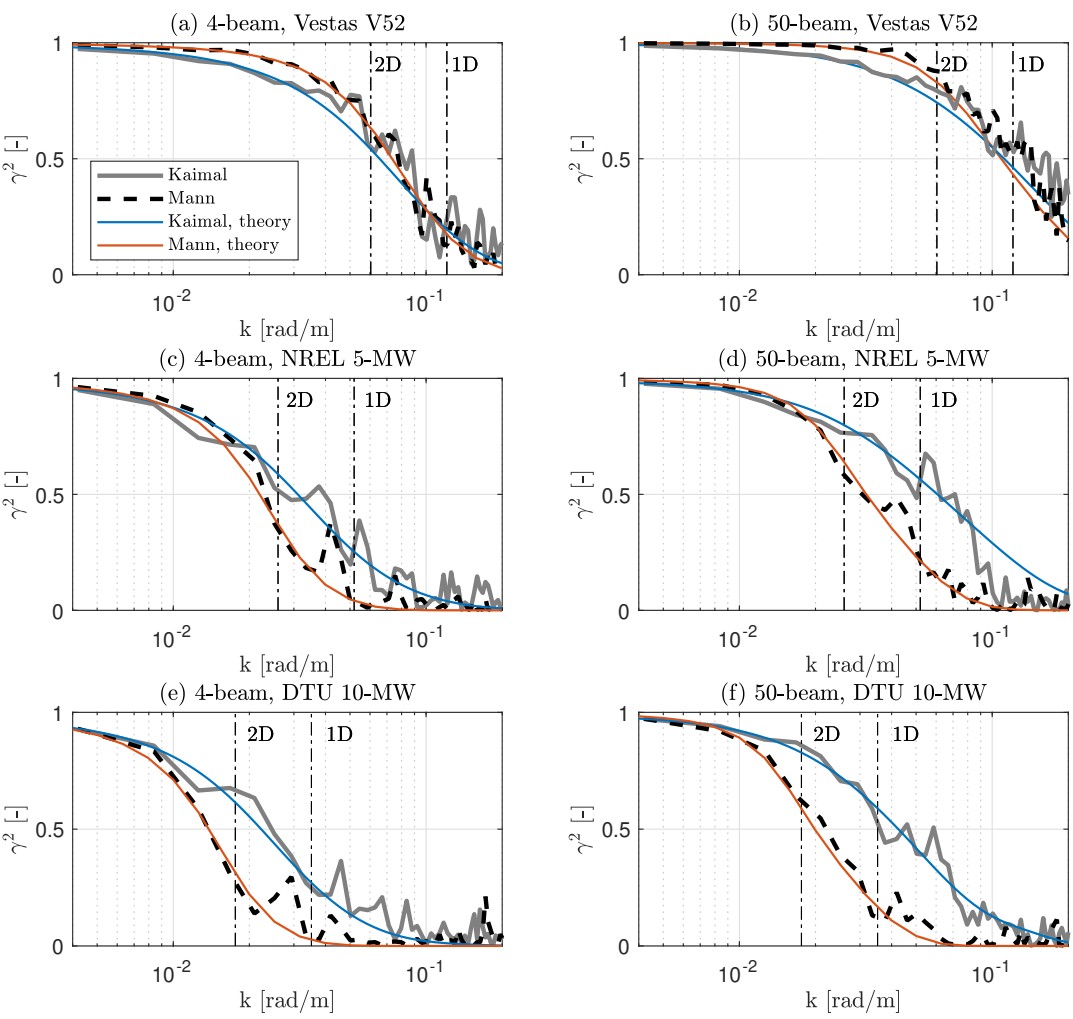

**Figure 3.** Magnitude-squared coherence $\gamma^2$ between LiDAR measurements and the rotor effective wind speed. The left column contains results for the 4-beam scan pattern. The right column represents the 50-beam circular scan pattern. From top to bottom, the plots show results for the Vestas V52 turbine, NREL 5-MW turbine and DTU 10-MW turbine. The dash-dot lines labeled $1D$ and $2D$ represent the wave numbers $k = \frac{2\pi}{D}$ and $k = \frac{2\pi}{2D}$, respectively. In the legend, "theory" denotes the theoretical coherence.





*Data availability.* The turbulence box data could be made available on request.

*Author contributions.* Liang Dong: Conceptualization, Methodology, Software, Investigation, Writing - original draft. Wai Hou Lio: Con-
ceptualization, Methodology, Investigation, Writing - original draft. Eric Simley: Methodology, Investigation, Writing - original draft.

*Competing interests.* The authors declare that they have no conflict of interest.

*Acknowledgements.* This research was supported by the Energy Technology Development and Demonstration Program (EUDP), LiDAR
assisted control for reliability improvement (LICOREIM, Grant No. 64019-0580).

The authors also want to thank Jakob Mann for providing the inputs of theoretical coherence results with the Mann model, and his
suggestions on the manuscript.

This work was authored in part by the National Renewable Energy Laboratory, operated by Alliance for Sustainable Energy, LLC, for
the U.S. Department of Energy (DOE) under Contract No. DE-AC36-08GO28308. Funding provided by the U.S. Department of Energy
Office of Energy Efficiency and Renewable Energy Wind Energy Technologies Office. The views expressed in the article do not necessarily
represent the views of the DOE or the U.S. Government. The U.S. Government retains and the publisher, by accepting the article for
publication, acknowledges that the U.S. Government retains a nonexclusive, paid-up, irrevocable, worldwide license to publish or reproduce
the published form of this work, or allow others to do so, for U.S. Government purposes.





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
