# Peer review of "On Turbulence Models and LiDAR Measurements for Wind Turbine Control"

_Wind Energy Science, 2021_

## Author Comment (AC1)

**Author Response to Reviews of**

**On Turbulence Models and LiDAR Measurements for Wind Turbine Control**

Liang Dong, Wai Hou Lio, Eric Simley
*Wind Energy Science Discuss,* `doi:10.5194/wes-2021-51`
* * *
**RC:** *Reviewer Comment*,     AR: *Author Response*,     ☐ Manuscript text

Dear Reviewers,

We sincerely appreciate all valuable comments and suggestions. We have revised the manuscript based on your comments. You will find our responses to your comments below.

**1.   Reviewer #1**

**RC:**   *The paper numerically investigates the spectral coherence between a simulated lidar measurement of the wind inflow to a wind turbine on the one hand, and the rotor-effective wind speed (REWS) at the turbine on the other hand. The incoming wind field is simulated by two different, commonly used turbulence models, namely the Kaimal and the Mann model. Results are evaluated for three different rotor diameters, namely 52 m, 126 m, and 178 m. It is found that the above-mentioned coherence significantly differs between the turbulence models as well as between the rotor diameters. Namely, the coherence decreases with the rotor size, and this decrease is stronger for the Mann model.*

**RC:**   *The methodology and investigations are based on results by Held and Mann (2019) where a significant difference in coherence was found between the same turbulence models. In that paper, only a 52 m rotor diameter was investigated, and additional validation against experimental field data was performed. As a consequence, it was found that the predictions of the Mann model came closer to reality than those of the Kaimal model.*

**RC:**   *The paper under review investigates a relevant and interesting topic, namely systematic differences between turbulence models, and their relevance for wind turbine applications. The numerical investigations are performed systematically and they deliver clear results. The main conclusion of the paper is that the systematic differences in coherence depending on the turbulence model should be considered as uncertainties for relevant applications.*

 AR:   *We appreciate your interest and valuable comments on this topic.*

**1.1.**

**RC:**   *One fundamental weakness of the paper is that turbulence models, including the two under investigation, differ significantly from real turbulence. Conclusions from the results of the paper for real-world applications are therefore very difficult and should be discussed with care. Other than in Held and Mann (2019), no experimental validation was performed here.*

 AR:   *According to wind turbine design requirement in IEC standards[1], for the standard wind turbine classes, the turbulence model must satisfy the three requirements: the turbulence standard deviation, the longitudinal*

*turbulence scale parameters, and a recognized model for the coherence. The random wind velocity field for the turbulence models shall satisfy the Kaimal model together with the coherence model described in Sec 2.1. As an alternative the Mann model can also be applied. So in this work, the Mann and the Kaimal model are selected for investigation, which is important for standard wind turbine design in wind energy industry.*

AR:   *The purpose of this work is not to analyze which turbulence model is more suitable for representing the real turbulence. Instead, this work focused on how to evaluate the benefit of LAC during the wind turbine design phase according to the IEC standards, and the importance of the turbulence model while the LAC is adopted is addressed.*

AR:   *The following section has been revised.*

> According to current wind turbine design requirements in the IEC standard [1], for the standard wind turbine classes the turbulence model must contain the following elements: the turbulence standard deviation, the longitudinal turbulence scale parameters, and a recognized model for the coherence. The standards recommend the use of either the Kaimal turbulence model, together with a standard exponential lateral-vertical coherence model, or the Mann turbulence model to represent the random wind velocity field. Although extensive research has been carried out on evaluating LiDAR measurement coherence, there is a clear knowledge gap regarding the impact of different turbulence models on the LiDAR measurement coherence. The wind field model used in most of the above-mentioned studies consists of the Kaimal turbulence spectrum and the lateral-vertical spatial coherence model defined in the IEC standard [1]. The impact of different turbulence models on the dynamic response of an offshore wind turbine has been evaluated by [2]; the results showed that as the rotor size becomes larger, the variation of the wind in time and space also becomes increasingly important. There is a need to evaluate the load reduction potential of LAC using different turbulence models, which is critical for determining the value creation of LAC during the wind turbine design phase. ...

AR:   *This work also suggests that the experimental validation is critical, since the atmospheric conditions are different at different site, the turbine design with LAC feature should consider this site-specific parameters. This suggestion is included in Section 5.*

**1.2.**

RC:   **The text of the paper is not strictly systematic. A more strict systematic would improve the text. Especially, conclusions are already drawn in section 4.3 (last paragraph), but they should be moved to section 5. In the conclusions themselves, these findings are missing. Moreover, two further important findings of the paper are completely missing:**

  - **The rotor size influences the difference in coherence between the turbulence models.**

  - **An experimental validation of the findings, especially for large rotors, is essential for further application.**

AR:   *Thanks for this suggestion. We have reorganized the conclusion accordingly.*

**1.3.**

**RC:** *A few inappropriate terms make the understanding of the unnecessarily difficult. Those are Lidar measurement quality (P1L23): This term is used frequently throughout the paper. It does make no sense at all in a simulation study, because the quality of the lidar measurement is not accessible and also not investigated here. This causes unnecessary confusion. What is meant (to my understanding) is the quality of the prediction of the REWS by the two turbulence models, given the information of a simulated lidar measurement upstream of the rotor. The term should be replaced by something more appropriate, like "REWS prediction quality" (this is probably not the best term either), and it should be clarified and explained when first used. In contrast to this, the term "lidar measurement coherence", which is also used frequently, does actually make sense, even though no real measurement is performed.*

**AR:** *Agreed, this term does cause some unnecessary confusions for readers. The "LiDAR measurement quality" is replaced by "simulated measurement quality metrics". The term is clarified and explained in detail as follows.*

> Three commonly used simulated measurement quality metrics for LAC application are defined in [3]: magnitude-squared coherence between the true rotor effective wind speed and the LiDAR-based estimate, mean square error (MSE) between the true rotor effective wind speed and the LiDAR-based estimate, and MSE between the generator speed and the rated generate speed.

**1.4.**

**RC:** *Value creation of LAC (P1L8): Without any relation the field measurements, it is not appropriate to speak of "value creation" in this context. There is no way to evaluate the real benefits of LAC based on the presented results.*

**AR:** *As mentioned in Section 1.1, the "value creation" means "the potential value creation of LAC based on simulations during the wind turbine design phase". The abstract has been revised.*

> ...In other words, the potential value creation of LAC based on simulations during the wind turbine design phase, evaluated using the Kaimal turbulence model, will be diminished if the Mann turbulence model is used instead. ...

**AR:** *According to the work [3], by optimizing the LiDAR scan pattern, the higher measurement coherence bandwidth can be achieved, but the cost of LiDAR will increase as well. Generally speaking, the lower the value of $k_{0.5}$, the lower the LAC benefits. Integrating LAC into the turbine design phase involves a trade-off optimization problem to consider the turbine cost and LiDAR cost simultaneously.*

**AR:** *The "value creation" is strongly dependent on the coherence bandwidth $k_{0.5}$ as described in Section 4.2. It is feasible to evaluate the benefits, for example, normal power production simulation according to IEC standards can be performed using reference wind turbines and aero-elastic tool HAWC2, which remains future work. This description has been added to the conclusions.*

> ... Note that the impacts on the load reduction need to be further investigated using reference turbines and aero-elastic tools following the IEC standards. ...

**1.5.**

**RC:** *Lidar measurement error (P5L138): It is not clear what this term means here. Error of what compared to what exactly?*

**AR:** *To be more clear, we have changed "Lidar measurement error" to the coherence between the rotor effective wind speed and its estimated value base on LiDAR measurement.*

> Otherwise, without the correlation of the $v$ and $w$ components the coherence between the REWS and its estimated value based on LiDAR measurements could be unrealistically high, because the contribution of the $v$ and $w$ components could be close to zero after spatial averaging along the LiDAR beams.

**1.6.**

**RC:** *Moreover, please make sure that all technical terms are clearly defined or explained at their first occurrence.*

**AR:** *We have checked all the technical terms to make sure they are explained at the first occurrence.*

**1.7.**

**RC:** *P7L159: "the choice of turbulence model strongly influences the coherence of LiDAR measurements". See above, "lidar measurement quality". Reformulate to what is actually meant.*

**AR:** *The description has been revised.*

> So the findings above indicate that the choice of turbulence model strongly influences the correlation between the Lidar measurement and true rotor effective wind speed.

**1.8.**

**RC:** *P9L178: "the true velocity measured by a lidar". What is meant by "true velocity"? Is is the LOS component a a certain point? Keep in mind that velocity is a vector by definition. Make clear.*

**AR:** *The "true velocity" means the volume measurement should be modeled using range weighting function for a real scanning LiDAR system. The description has been revised.*

> The velocity measured by a real scanning LiDAR is a spatial average of the LOS velocities along the LiDAR beam, which is described by the range weighting function.

**1.9.**

**RC:** *Section 4.1: Was the lidar measurement modeled after section 3.2? Was the REWS evaluated after section 3.3? What are the time constants in the measurement?*

**AR:** *Yes, the method of simulating LiDAR measurements is described in Section 3.2, and the method of calculating REWS is in Section 3.3.*

AR:   *I presume the "time constants" means the LiDAR scanning frequency. I have added a scanning frequency and a LOS measurement frequency in Table 2.*

> ... For both LiDAR scan patterns, the scanning frequency for completing a full scan is 1 Hz, and the LOS measurement frequency is 4 Hz and 50 Hz based on commercial examples.

**1.10.**

RC:   *Section 4.2: The discussion of eddy sizes (line 219 ff) is confusing. First, the mentioned integral length scale is questionable and will most probably depend strongly on the time window used for the analysis. Moreover, what is meant by "the eddy size" is probably "the size of the largest eddies", which in turn is a questionable quantity. A spherical eddy is hard to imagine. Moreover, if it would exist, pitching the blades to "feather" would not help in decreasing the loads. These aspects are, however, unnecessary for the relevant part of the discussion. Namely that, given Taylor's hypothesis of frozen turbulence, the smallest relevant time and length scale for collective pitch control is of the order of the rotor diameter. The authors should restrict the discussion to this aspect.*

AR:   *Thanks for the comment. We agreed that the "$3D$ spherical eddy" is not clear and it makes the explanation more complicated. The irrelevant term has been removed. So here we revised the description and focused on the $1D$-diameter eddy across the rotor. And a new reference [4] is cited here to help justify the argument.*

AR:   *Regarding the description "pitch to feather", we agreed that to reduce the fatigue loads using LAC, the pitch should follow the wind variation, not just "pitch to feather". So, "pitch to feather" has been removed.*

> For reducing fatigue loads using LAC, detecting eddies with a length as small as $1D$ in the longitudinal direction is important, because the thrust load induced by eddies with diameters of $1D$ or larger across the rotor in the lateral and vertical directions can be mitigated using collective pitch control [4]; in turn, eddies covering the entire rotor disc in the lateral and vertical directions are expected to extend at least $1D$ in the longitudinal direction. Thus, the magnitude-squared coherence $\gamma^2$ at $k = 2\pi/D$ is the most critical metric.

**2.   Reviewer #2**

**2.1.**

RC:   *In this paper, the authors compare two different turbulence models (the Kaimal model and the Mann model) and two different LiDAR scanning protocols to assess their potential for LiDAR assisted control (LAC). The focus of this work is on larger rotor sizes. The comparisons are based on numerical implementations of the turbulence models and simulations of the LiDAR scanning protocols. The authors find a good agreement of the magnitude-squared coherence between simulated LiDAR measurements and the rotor effective wind speed with the theoretically expected ones.*

RC:   *Overall, the work is interesting, clearly structured, and well written. The main finding is that the asserted coherence depends strongly on the specific type of turbulence model. In fact, it is shown that the assertion of the benefit for LAC depends more on the turbulence model than on the specific LiDAR scanning pattern.*

AR:   *We appreciate your interest and valuable comments on this topic.*

**2.2.**

RC:   ***This brings me to my main question. Whether LAC is beneficial or not ultimately depends on the co-herence of the atmospheric turbulence. This means that the major question should be which turbulence model is better suited to capture atmospheric turbulence. In that sense, sentences like "the value creation of LAC, evaluated using the Kaimal turbulence model, will be diminished if the Mann turbulence model is used instead." seem inappropriate to me since no direct conclusions for field measurements can be drawn. I think the authors should clarify their scope and in particular the limitations of their study.***

AR:   *Thanks for the comment. This comment is similar to the comment Section 1.1.*

AR:   *As mentioned in comment Section 1.1, the Mann and the Kaimal model are selected for investigation, which is important for standard wind turbine design in wind energy industry. The purpose of this work is not to analyze which turbulence model is more suitable for representing the real turbulence. Instead, this work focused on how to evaluate the benefit of LAC during the wind turbine design phase according to IEC standards, and the importance of the turbulence model while the LAC is adopted is addressed. This scope has been clarified as shown in Section 1.1.*

AR:   *The potential benefits of LAC is highly dependent on the magnitude-squared coherence. Low coherence will result in low benefits as discussed in the manuscript Section 4.2.*

**2.3.**

RC:   ***Another question concerns the choice of parameters. Many of the parameters in this study are kept fixed, which makes me wonder how much the conclusions depend on the specific parameter choices. Can the authors comment on that?***

AR:   *The parameters of turbulence model is chosen according to IEC standard, which is commonly used for standard wind turbine design, and fit with the scope of this work.*

AR:   *The parameters of LiDAR optimal scanning pattern are chosen according to the previous work [3].*

**2.4.**

RC:   ***Eq. (5): The argument of $R_{ij}$ should be boldfaced.***

AR:   *Eq. (5) has been modified.*

$$R_{ij}(\mathbf{r}) = \langle u_i(\mathbf{x}) u_j(\mathbf{x} + \mathbf{r}) \rangle, \tag{1}$$

**2.5.**

RC:   ***I don't understand the statement below eq. (7): "When the two indices $i = j$, then $\Delta_y = \Delta_x = 0$..." Can't I vary $i$, $j$, $\Delta_x$, and $\Delta_y$ independently? I think this statement needs clarification.***

AR:   *For each pair of points $i$ and $j$, there should be a specific separation in the x and y directions. So changing the indices will determine the $\Delta_x$ and $\Delta_y$.*

AR: *The statement is used to illustrate the auto-spectrum $\Psi_{ii}(k_1)$ and $\Psi_{jj}(k_1)$ in the denominator. The description has been revised.*

> For the denominator in Eq.(7), when the two indices $i = j$, then $\Delta_y = \Delta_z = 0$ and the wave number auto-spectrum $\Psi_{ii}(k_1)$ and $\Psi_{jj}(k_1)$ are expressed as
>
> $$\Psi_{xx}(k_1) = \int_{-\infty}^{\infty} \int_{-\infty}^{\infty} \Phi_{xx}(\mathbf{k}) dk_2 dk_3, \tag{2}$$
>
> where the subscript $xx \in [ii, jj]$.

**2.6.**

RC: *Line 114: I think "the" in "especially for the Mann model" should be deleted.*

AR: *"the" is deleted.*

**2.7.**

RC: *Eq. (9): The notation is a bit difficult to understand: Shouldn't all vectors be boldfaced? Perhaps it is worth checking the manuscript once more regarding the consistency of notation.*

AR: *To make it more clear, the description has been revised as follows.*

[revised manuscript text omitted]